A pilot study to investigate if New Zealand men with prostate cancer benefit from a Mediterranean-style diet

Erdrich Sharon 1 sharon@houseofhealth.co.nz
Bishop Karen S. 2
Karunasinghe Nishi 2
Han Dug Yeo 3
Ferguson Lynnette R. 1 2 3
1 Discipline of Nutrition, FM&HS, University of Auckland , Auckland , New Zealand
2 Auckland Cancer Society Research Centre, FM & HS, University of Auckland , Auckland , New Zealand
3 Nutrigenomics New Zealand, University of Auckland , Auckland , New Zealand
Sanderson J Thomas
Electronic publication date: 2015 Jul 2
Publication date: 2015
Volume: 3
Electronic Location ID: e1080
Received 2014 Dec 20; Accepted 2015 Jun 14
Copyright: © 2015 Erdrich et al.
Copyright year: 2015
Copyright holder: Erdrich et al.
License: This is an open access article distributed under the terms of the Creative Commons Attribution License, which permits unrestricted use, distribution, reproduction and adaptation in any medium and for any purpose provided that it is properly attributed. For attribution, the original author(s), title, publication source (PeerJ) and either DOI or URL of the article must be cited.
License URL: https://creativecommons.org/licenses/by/4.0/

Keywords: Antioxidants, DNA damage, Mediterranean style diet, Prostate cancer, Nutrition

Funding: Graduate Student Fund, from the University of Auckland Funding was received from the Auckland Cancer Research Society. The funders had no role in study design, data collection and analysis, decision to publish, or preparation of the manuscript.

==============================
Carcinoma of the prostate is the most commonly diagnosed malignancy and the third leading cause of mortality in New Zealand men, making it a significant health issue in this country. Global distribution patterns suggest that diet and lifestyle factors may be linked to the development and progression of this cancer. Twenty men with diagnosed prostate cancer adhered to a Mediterranean diet, with specific adaptations, for three months. Prostate-specific antigen, C-reactive protein and DNA damage were evaluated at baseline and after three months of following the diet. Dietary data were collated from diet diaries and an adaptation of a validated Mediterranean diet questionnaire. A significant reduction in DNA damage compared to baseline was apparent, with particular benefit noted for overall adherence to the diet (p = 0.013), increased intake of folate (p = 0.023), vitamin C (p = 0.007), legumes (p = 0.004) and green tea (p = 0.002). Higher intakes of red meat and dairy products were inversely associated with DNA damage (p = 0.003 and p = 0.008 respectively). The results from this small feasibility study suggest that a high-antioxidant diet, modelled on Mediterranean traditions, may be of benefit for men with prostate cancer. Protection against DNA damage appears to be associated with the diet implemented, ostensibly due to reduction in reactive oxidant species. These findings warrant further exploration in a longer trial, with a larger cohort.

Introduction

On a global scale, prostate cancer is an important health consideration. It is the fourth most common cancer internationally, and in men ranks second only to lung cancer (Ferlay et al., 2012). Prostate cancer incidence is highest in developed nations, compared to less-developed countries, a difference that is only partially explained by the higher use of prostate specific antigen (PSA) as a screening tool in developed nations (Center et al., 2012). In New Zealand, cancer of the prostate is the third most common cause of cancer-related mortality, with death rates disproportionately higher in Māori men (28.7 c.f. 16.7 deaths per 100,000 (age-standardised rates per 100,000) male population, standardised to the WHO world standard population) (MOH, 2013).

Worldwide patterns of prostate cancer incidence and mortality support the hypothesis that diet and lifestyle are likely contributors to both development and progression of this malignancy. Furthermore, alterations in risk associated with migratory patterns and the westernisation of dietary patterns associated with globalisation (Baade, Youlden & Krnjacki, 2009; Melnik, John & Schmitz, 2011) give added credence to this theory.

Chronic inflammation and infection have been implicated in the development of around one-fifth of all cancers (Greene et al., 2011), including prostate cancer (Gurel et al., 2014). Other influences include genetic and epigenetic factors (Pantuck et al., 2006), imbalances between reactive oxygen species and antioxidants, and DNA damage (Waris & Ahsan, 2006) (Fig. 1).

Figure 1 Factors involved in the pathogenesis of tumour development (Pantuck et al., 2006; Waris & Ahsan, 2006).

The role of inflammation in prostate cancer is unclear. Elevated levels of inflammatory markers have been associated with high-grade prostate cancer in some studies (Shariat et al., 2001; Platz & De Marzo, 2004; Huffman et al., 2006; Gurel et al., 2014) but not in others (Il’yasova et al., 2005; Stark et al., 2009). Systemic inflammation has been associated with earlier cancer mortality (McArdle, Qayyum & McMillan, 2010; Shafique et al., 2012), which adds weight to an argument for a detrimental effect of chronic inflammation and the potential benefit of alow-inflammatory diet.

Increased levels of free radicals and pro-oxidant compounds (Arsova-Sarafinovska et al., 2009; Qu et al., 2013; Kanwal et al., 2014), and decreased levels of antioxidant enzymes have been found in prostate tissue (Kanwal et al., 2014), and in association with prostate cancer (Arsova-Sarafinovska et al., 2009). Both prostatic cancer cells and high-grade prostatic intraepithelial neoplasia are notably deficient in the important endogenous antioxidant, glutathione S-transferase (Platz & De Marzo, 2004). This may be due to inflammation-induced inactivation of genes that have roles in cellular protection and restoration of damaged DNA (Kundu & Surh, 2012). Reduction in the ability of cells to produce protective antioxidants may occur due to alterations in cell morphology, leaving the prostate vulnerable to damage by carcinogenic compounds that might otherwise be neutralised by antioxidants. Indeed, proliferative inflammatory atrophy, a precursor to development of prostate cancer, is a cellular change that is postulated to be the result of cell damage (Platz & De Marzo, 2004; Brawer, 2005).

The Mediterranean diet has been extensively examined and its benefits in terms of reductions in oxidative stress and inflammation are generally well-accepted (Urpi-Sarda et al., 2012; Viscogliosi et al., 2013). This dietary style is centered on consumption of high amounts of extra virgin olive oil, fruit and vegetables, pulses and legumes, whole-grains, and poultry, along with some fish and seafood. Intake of red meat, dairy products and processed or refined foods is traditionally low (Couto et al., 2011). Deviation away from such a pattern, towards a more western-style diet, has been associated with increased prostate cancer incidence (Ambrosini et al., 2008; Stott-Miller, Neuhouser & Stanford, 2013). Couto et al. (2011) examined data from the large European Prospective Investigation into Cancer and Nutrition (EPIC) study and concluded that a Mediterranean diet is particularly beneficial in protecting against breast, colon, and prostate cancer.

The brassica family is not emphasised in general Mediterranean diet guidelines. However, this family of vegetables has attracted much interest in recent years, primarily on account of the levels of glucosinolates, which are particularly high in broccoli (Moreno et al., 2006). Nutrigenomic effects of broccoli are discussed in detail in Ferguson & Schlothauer (2012).

Pomegranate, and its juice, has received a great deal of attention related to potential chemo-protective effects, including beneficial effects in slowing of prostate-specific antigen (PSA) doubling time in men with prostate cancer (Pantuck et al., 2006; Pantuck et al., 2009; Paller et al., 2012). This benefit is attributed to high levels of polyphenols (particularly punicalagin, an elligatannin (Koyama et al., 2010)), that contribute to pomegranate’s overall antioxidant capacity, which is greater than either red wine or green tea (Gil et al., 2000). Polyphenols are also high in extra virgin olive oil, red wine, and green tea (Tuck & Hayball, 2002). These phytochemicals have demonstrable epigenetic effects (Joven et al., 2013), which may account for at least some of the benefits attributed to their consumption. As pomegranate is found throughout the Mediterranean area, it is logical to assume it would be commonly consumed in the region. However, this has not been reported in nutritional research which focuses on Mediterranean dietary patterns. It is quite possible that pomegranate may be a contributor to documented advantages associated with diets in the Mediterranean region.

High fibre diets, including consumption of legumes and whole grains, have been linked to a wide range of health benefits. Legumes are an important food group in the Mediterranean diet (Ferrís-Tortajada et al., 2012), and have been associated with reduced prostate cancer risk (Chan, Lok & Woo, 2009). Consumption of legumes (particularly soy) is significant in Asian diets where prostate cancer incidence is also low (Chan, Lok & Woo, 2009). Mechanisms for benefit are numerous and include: high fibre content, which may be advantageous by reduction of post-prandial glycaemia (Gropper, Stepnick & Smith, 2013), lower levels of insulin-like growth factor (Landberg et al., 2010), and increases in sex hormone binding globulin (Tymchuk et al., 1998). The minimisation of post-prandial rises in blood sugar level is desirable in men with prostate cancer, due to glycaemia-associated increases in markers of inflammation and oxidative stress (Rytter et al., 2009). Additionally, foods that promote a high glycaemic response induce insulin and insulin-like growth factor (Melnik, John & Schmitz, 2011), which may contribute to prostate cancer progression (Chan et al., 2002).

In some studies (Gao, LaValley & Tucker, 2005; Koh et al., 2006; Allen et al., 2008), but not all (Huncharek, Muscat & Kupelnick, 2008; Pettersson et al., 2012), consumption of dairy foods has been linked to higher risk of developing prostate cancer. Risk has been associated with the quantity consumed (Allen et al., 2008) and dairy intake during adolescence (Torfadottir et al., 2012). Prostate cancer mortality has also been correlated to milk consumption (Ganmaa et al., 2002). While these data are inconclusive, it does, nonetheless, raise concerns over the recommendation of dairy as a source of calcium for men at risk of developing osteoporosis, including those with prostate cancer who have had hormonal ablation treatment (Malcolm et al., 2007).

Fish is not considered a major component of Mediterranean diets (Trichopoulou et al., 2005), but is generally considered preferential to red meat for men with prostate cancer (Terry et al., 2001; Chavarro et al., 2008; Bosire et al., 2013). Fish is regarded as a good source of omega-3 polyunsaturated fatty acids, promoting anti-inflammatory pathways (Jain et al., 2008), which has been considered to be desirable in men for whom lowering inflammation is a goal. However, the benefit of omega-3 polyunsaturated fatty acids in prostate cancer has recently been challenged (Brasky et al., 2013). The role of dietary fats in prostate cancer is discussed in more detail by Bishop et al. (2015).

Green tea is not a dietary feature in the Mediterranean region but is a common beverage in East Asian countries, where mortality rates from prostate cancer are the lowest, globally (Ferlay et al., 2012). A mounting volume of evidence supports the recommendation of consumption of green tea, due to its antioxidant potential, largely from polyphenols, in particular epigallocatechin-3-gallate (EGCG) (Du et al., 2012). EGCG has documented anti-proliferative properties (Du et al., 2012) and affords protection to DNA in prostate cells (Kanwal et al., 2014). Furthermore, green tea consumption has been associated with lower prostate cancer incidence (Zheng et al., 2011) and reduced risk of progression to advanced disease (Kurahashi et al., 2008).

This pilot study was undertaken to establish both feasibility and likely benefit of three months of adherence to a Mediterranean dietary pattern, with specific modifications, on DNA damage and inflammation in New Zealand men with prostate cancer.

Materials and Methods

Written approval for this pilot study was granted by the Northern Y Regional Ethics Committee (New Zealand), study reference NTY/11/11/109.

Study volunteers with Gleason score 6—7 (3 + 3 or 3 + 4) who had participated in an earlier study investigating DNA damage and genotypes (Karunasinghe et al., 2012) were invited to enrol in this dietary intervention. Men with lower Gleason scores were selected preferentially to minimise the possible confounding effect of advanced disease. Other criteria for inclusion were: under 75 years of age, no diagnosis of diabetes, no evidence of progression and not currently receiving treatment for prostate cancer (hormonal therapy excepted), and with no history of other cancer except treated skin carcinomas.

The dietary guidelines were in accordance with general Mediterranean patterns as per a validated tool (Martinez-Gonzalez et al., 2012) with specific adaptations.

Men were given nutritional counselling at enrolment by nutritional specialists on the study team and coached regarding the dietary guidelines. Both hard and soft copies of a comprehensive compilation of recipes incorporating the main foods and principles of the diet were provided. Additional support and clarification was provided on an as-needed basis. As a goal of this study was to determine the degree of acceptance of this pattern of eating in the lifestyle of New Zealand men, the intervention was based around general principles with specific recommendations, as summarised in Table 1, rather than a prescriptive diet.

Table 1 A brief outline of food items/groups that study-participants were asked to consume or avoid.

Foods to consume	Recommendationa	Foods to avoid/limit	Quantity	
Oily fish	≥2 servings weekly	Red meat	≤1 serving weekly	
Olive oil	≥ l tblsp daily	Butter (high fat dairy)	Avoid/reduce	
Chicken (skinless)	≤2 servings weekly	Processed meats	Avoid	
Fresh vegetables (broccolib, dark leafy vegetables, cooked tomatoes, sweet potato & salads)	≥5 servings daily	Eggs	1–4 per week	
Whole grains	Daily as required	Fruit (berriesb)	Up to 2 portions daily	
Red wine	≤2 glasses daily	Refined foods (white bread, crisps, pies)	Reduce or avoid	
Green tea	≥2 cups daily	Other alcohol	Avoid	
Pomegranate juice	150 mL daily	Fruit juice	Avoid	
Fats and oils (avocado, olives, nuts and seeds)	Daily as required	Sugar (biscuits, cakes, sweets)	Avoid	
Notes.

a Unless contraindicated due to health problems, allergies or intolerances.

b Preferred items.

All participants were provided with New Zealand-produced extra virgin olive oil with a polyphenol level of 233 mg/kg (1 L/month), fresh frozen salmon (200 g/week), unsweetened pure pomegranate juice (1 L/week) and samples of a variety of canned legumes. These were donated by New Zealand distributors: Oil Seed Extractions Ltd., Ashburton; Aoraki Smokehouse Salmon, Twizel; Life Juices, Auckland, and Delmaine Fine Foods, Auckland, respectively, and given to the volunteers at enrolment.

The validated 14-point Mediterranean diet adherence questionnaire published by Martinez-Gonzalez et al. (2012) was adapted to be consistent with the recommendations. The questionnaire was completed by study participants, along with four-day diet diaries, at the beginning and conclusion of the study (Table 2). Adherence questionnaires were scored out of 20. Serving sizes were estimated and recorded both in diet diaries and the dietary adherence questionnaire. Diet diaries were analysed using Food Works® 7 (Xyris 2012, Xyris Software (Australia) Pty Ltd, Brisbane, Australia).

Table 2 Dietary Adherence Questionnaire.

Adapted from Martinez-Gonzalez et al. (2012).

Assessing your “Mediterranean Diet”	
Please indicate your answers to the questions in the space provided.	
Question	Answer	
1. Do you use olive oil as a culinary fat?		
2. How much olive oil do you use in a given day (including oil used for frying, salads, out-of-house meals, etc.)?		
3. How many vegetable servings do you consume per day? (1 serving: 1/2 cup cooked vegetables or 1 cup raw vegetables) (consider side-dishes as half a serving) Cooked:Raw/salad:		
4. How many fruit units (including natural fruit juices) do you consume per day?		
5. How many servings of red meat, hamburger, or meat products (ham, sausage, etc.) do you consume per week? (1 serving: 100–150 g)		
6. How many servings of butter, margarine or cream do you consume per day? (1 serving: 12 g/approx. 2 tsp)		
7. How many sweet or carbonated beverages do you drink per day?		
8. How much wine do you drink per week?		
9. How many servings of legumes do you consume per week? (1 serving: 150 g)		
10. How many servings of fish or shellfish do you consume per week? (1 serving: 100–150 g of fish, or 200 g/4–5 units of shellfish)		
11. How many times per week do you consume sweets or pastries, such as cakes, cookies, biscuits or custard?		
12. How many servings of nuts (including peanuts) do you consume per week? (1 serving: 30 g)		
13. Do you preferentially consume chicken, turkey or rabbit meat instead of veal, pork, hamburger or sausage?		
14. How many times per week do you consume vegetables, pasta, rice or other dishes seasoned with sofrito (sauce made with tomato and onion, leek or garlic and simmered with olive oil)?		
15. How many servings of pomegranate fruit (1 serving: 1 piece), pomegranate juice (1 serving: 150 mL), or of pomegranate molasses (1 serving: 1/2 tblsp) do you have per day?		
16. How much alcohol (other than wine) do you drink per week? (1 serving: 1 nip spirits, 300 mL beer)		
17. How many cups of green tea do you drink per day?		
18. How many servings of broccoli do you consume per week?		
19. How many servings of dairy products do you have per week? (1 serving: 30 g cheese, 250 mL milk, 100 mL yoghurt)		
20. Do you preferentially use wholegrain bread and crackers instead of white/refined bread and crackers?		

Blood samples were collected at baseline, and again at three months of follow-up, (plain, EDTA, Heparin and SST II Advance Vaccutainers), and processed within 2 h of collection. C-reactive protein and PSA were tested using enzyme immunoassays performed by LabTests, Auckland, New Zealand.

Comet assays were performed on both fresh blood and hydrogen-peroxide challenged samples as described by Karunasinghe et al. (2004) and Ferguson et al. (2010). As the data were skewed, figures for percentage tail DNA were log-transformed and the back-transformed mean was used for the analysis.

Statistical evaluation was performed using SAS (v9.2 SAS Institute, Cary, NC, USA). Correlations were examined using Spearman’s rho. The Student’s paired t-Test was used to evaluate differences between baseline and study-end.

Results

A total of twenty-eight men were enrolled in the study. While we sought to enrol men on active surveillance protocols, a low response rate resulted in the recruitment of those who had received treatment for PCa but within the Gleason score range specified. Eight men were not included in the final analysis: four were lost to follow-up for a variety of reasons, two dropped out of the study due to life stresses, one had difficulty conforming to the diet and one was eliminated due to unreliability of dietary information provided. The characteristics for the remaining twenty participants are presented in Table 3.

Table 3 Characteristics of study participants at baseline.

Characteristic	n	Characteristic	n	
Age 52–74 Years		BMI 23–33 kg/m2		
50–59	3	≥20–≤ 25	4	
60–69	12	>25–≤30	12	
>70	5	>30	4	
Marital status		Family history of PCa		
Married or with partner	17	1° relative with PCa	3	
Single or widowed	3	Other relative with PCa	2	
Ethnicity		Gleason score		
Caucasian	20	3 + 3	14	
Smoking status		3 + 4		
Never	7	Time since diagnosis (years)		
Former	13	Treatment type		
Current	0	None	6	
Activity Level a		Prostatectomy	10	
Heavy	1	Prostatectomy + Hormones + DxR	1	
Moderate	2	Prostatectomy + DxR	1	
Light-moderate	1	Hormones + DxR	1	
Light	5	Brachytherapy	1	
Sedentary	10	Medication		
Very sedentary	1	Statins	6	
Alcohol intake		Aspirin	5	
Non-drinkers	4	Diclofenac	4	
≤2 Standard drinks per day	10	Anti-hypertensive	4	
>2 Standard drinks per day	6	TNF-antagonist	1	
PSA (mean, μg/L)	1.53	Allopurinol	1	
CRP (mean, mg/L)	1.55	Other	6	
Notes.

BMI body mass index

1° first degree

PCa prostate cancer

DxR radiotherapy

PSA prostate-specific antigen

CRP C-reactive protein

a Physical activity level as defined by FoodWorks® 7.

Data on tumour staging was available for half of the cohort. Using the Tumour (T), Node (N), Metastasis (M) grading system, grades were T1–T3, N0 and M0.

Of the twenty participants in the final analysis, none were current smokers. Mean time since smoking cessation was 31 years (SD = 13). Past history of smoking of less than one pack-year was regarded as a “never-smoker”. Pack-year history was positively associated with the use of aspirin and/or diclofenac (p = 0.007).

Over the course of the study two men ceased taking and one commenced low dose aspirin. Another participant discontinued use of diclofenac during the study period.

Of this somewhat sedentary group of men, 70% were overweight or obese (BMI >25 kg/m2). Mean body weight reduced by 2.3 kg (95% CI [1.11–3.49], p < 0.001) over the course of the study. There was a mean reduction in body mass index of 0.85 kg/m2 (95% CI [0.52–1.18], p < 0.001).

Men who were less active tended to have higher levels of C-reactive protein (p = 0.003).This association remained at study end, albeit slightly weaker (p = 0.055).

At baseline, dietary scores for the targeted Mediterranean-style pattern were low. Mean adherence was 6.3 (SE 0.69), with individual scores ranging from 2 to 13 (of a maximum of 20). At three months of follow-up, mean adherence was 13.63 (SE 0.49), range 9–17. The mean change in dietary adherence from baseline to study end was +7.33 (95% CI [5.76–8.89]), which was highly significant (p < 0.001).

There were no statistically significant relationships between dietary adherence and either C-reactive protein or PSA at either baseline or three months.

Improvements were noted in all areas evaluated on the adherence questionnaire, with the exception of servings of fruit, vegetables, use of sofrito (tomato-based sauce prepared with garlic and/or onion), the intake of sweetened beverages, and wine. Pooled group adherence scores are presented in Table 4.

Table 4 Pooled dietary adherence scores from 20 participants at baseline and three months.

A maximum of 1 point for each item per participant was possible. 20 points reflects complete adherence by the whole cohort.

Dietary component	Criteria for one point	Baseline	Three months a	p	
1. Olive oil as culinary fat	Yes	13	20.0	0.021	
2. Olive oil used daily	≥1 tblsp	8	17.0	0.003	
3. Servings of vegetables/day	≥4	6	8.5	0.0563	
4. Servings of fruit/day (incl. pomegranate)	≤2	9	11.5	0.449	
5. Servings of red meat, hamburger, etc./week	<1	1	11.5	<0.001	
6. Servings of butter, margarine or cream/day	<1	3	14.5	<0.001	
7. Sweet or carbonated beverages/day	<1	16	20.0	0.083	
8. Wine consumed/week (glasses)	7–14	5	5.5	0.577	
9. Servings of legumes/week	≥5	6	16.0	<0.001	
10. Servings of fish or shellfish/week	≥3	4	14.0	<0.001	
11. Sweets or pastries/week	<3	7	16.0	<0.001	
12. Servings of nuts/week	≥5	5	16.0	<0.001	
13. Preference for chicken, etc.	Yes	7	18.0	<0.001	
14. Use of Sofrito sauce/week	≥2	6	9.5	0.297	
15. Servings of pomegranate/day	≥1	1	19.5	<0.001	
16. Units of other alcohol (excl. wine)/week	0	5	7.5	0.025	
17. Cups of green tea/day	≥2	2	9.0	0.008	
18. Servings of broccoli/week	≥5	0	6.5	0.004	
19. Servings of dairy products/week	≤5	5	13.0	0.003	
20. Use of whole grains	Yes	16	20.0	0.042	
Notes.

tblsp tablespoon

incl including

excl excluding

a Half-points were allocated wherever a shift towards improved adherence of ≥30% was evident.

Estimated energy requirements and reported energy intakewere calculated using FoodWorks® 7 software, from recorded body weights, reported energy expenditure and diet diaries (Table 5). There was a tendency to under-report energy intake. This was not statistically significant at baseline (SE 280, 95% CI [108–1063], p = 0.10). At the end of the study period, the difference reached statistical significance, with reported energy intake a mean 720 kilojoules lower than estimated requirements (SE 194, 95% CI [314–1125], p = 0.007).

Table 5 Comparison of estimated energy requirements and reported energy intake.

Energy intake	Baseline mean (SE)	Three months mean (SE)	Mean difference (95% CI)	p-value	
EER/kJ	10,675.6 (305.43)	10,476.05 (292.00)	−199.6 (−427.40–28.27)	0.083	
Reported energy intake/kJ	10,197.7 (341.64)	9,756.43 (267.09)	−441.3 (−970.60–88.05)	0.098	
Notes.

EER estimated energy requirement

kJ kilojoules

SE standard error

CI confidence interval

Energy obtained from saturated fat, as % of total energy intake, decreased significantly (p < 0.001). Other sources of energy did not alter significantly over the course of the study (Fig. 2).

Figure 2 Change in sources of energy expressed as % of total energy intake at baseline and three months.

SatFat = saturated fat; ∗p < 0.001.

Increases in intake of broccoli, sofrito, and pomegranate juice were statistically significant, as was a decrease in refined carbohydrate intake (per reported intakes of sweetened beverages and baked goods) (Table 6). The reduction in carbohydrate intake was not significant. No change was observed in regards to intake of fruit, vegetables, dietary fibre, or total sugar. It was apparent that the source of dietary sugars shifted away from sucrose and lactose, towards fructose and glucose.

Table 6 Changes in intake of dietary items and nutrients, from baseline to three months.

Dietary component	Baseline mean (SE)	3 months mean (SE)	Mean difference (95% CI)	p	
Carbohydrate (total) (g/day)	246.53 (11.20)	234.91 (11.00)	−11.63 (−30.47–7.22)	0.212	
Dietary fibre (total) (g/day)	31.23 (1.86)	32.28 (1.60)	1.04 (−1.83–3.92)	0.456	
Sugar (total) (g/day)	108.14 (7.90)	110.86 (7.96)	2.72 (−11.32–16.75)	0.690	
Glucose (g/day)	19.00 (2.28)	32.30 (2.97)	13.22 (7.30–19.20)	<0.001	
Fructose (g/day)	20.80 (2.40)	28.80 (2.80)	8.55 (2.80–14.30)	0.006	
Sucrose (g/day)	35.30 (4.80)	24.8 (3.50)	−10.56 (−17.75–1.37)	0.026	
Lactose (g/day)	12.80 (1.53)	6.53 (1.17)	−6.28 (−9.70–2.80)	0.001	
Folate (total) (μg/day)	537.00 (43.50)	564.00 (40.40)	27.00 (−31.00–85.40)	0.340	
Vitamin C (mg/day)	133.60 (13.20)	169.40 (22.40)	35.90 (−1.03–72.80)	0.056	
Vitamin E (mg/day)	18.60 (4.90)	26.53 (5.46)	7.94 (3.18–12.70)	0.005	
Vegetables (servings/day)	2.80 (0.28)	2.63(0.31)	−0.18(−1.13–0.78)	0.705	
Broccoli (servings/week)	1.58 (0.27)	2.42 (0.45)	0.84 (−0.19–1.49)	0.014	
Sofrito sauce (servings/week)	1.53 (0.45)	2.40 (0.47)	0.88 (0.28–1.47)	0.006	
Fruita (servings/day)	2.78 (0.49)	2.50 (0.31)	−0.28 (−1.01–0.46)	0.440	
Pomegranate (servings/day)	0.05 (0.05)	1.28 (0.16)	1.23 (0.86–1.59)	<0.001	
Sweetened beverages (servings/week)	0.51 (0.20)	0.15 (0.06)	−0.35 (−0.70–0.01)	0.046	
Cakes and biscuits (servings/week)	4.38 (1.01)	2.05 (0.55)	−2.33 (−3.82–0.83)	0.004	
Green tea (cups/day)	0.35 (0.17)	1.13 (0.30)	0.78 (−0.28–1.84)	0.004	
Notes.

g grams

SE standard error

CI confidence interval;

a All fruit and fruit juice, including pomegranate.

Participants significantly reduced their consumption of red meat (p < 0.001), and increased their intake of fish (p < 0.001), and legumes (p = 0.005), with no net change in protein intake (Table 7).

Table 7 Changes in protein intake from baseline to three months.

Dietary component	Baseline mean (SE)	3 months mean (SE)	Mean difference (95% CI)	p	
Protein (g/day)	106.73 (5.52)	99.49 (4.99)	−7.24 (−17.32–2.85)	0.149	
Red & processed meat (servings/week)	3.89 (0.48)	1.94 (0.36)	−1.95 (−2.59–1.32)	<0.001	
Fish (servings/week)	1.65 (0.20)	3.48 (0.46)	1.83 (0.91–2.74)	<0.001	
Legumes (servings/week)	2.37 (0.58)	3.78 (0.46)	1.41 (0.48–2.34)	0.005	
Notes.

SE standard error

CI confidence interval

Alterations in dietary fats and associated relationships are discussed in detail in Bishop et al. (2015).

Reductions in DNA damage were noted after three months of the dietary intervention. This did not reach significance for basal (fresh-blood) DNA damage (p = 0.075), but was highly significant for peroxide-induced DNA damage (p = 0.009).

Spearman bivariate correlation was used to identify relationships between DNA damage at study end and intake of the items specified on the adherence questionnaire and data generated from diet diaries. Overall, following the dietary pattern was inversely associated with DNA damage (p = 0.013). DNA damage was also inversely associated with consumption of green tea and intake of legumes (p = 0.002 and p = 0.004 respectively), and positively associated with red meat intake (p = 0.007). Intake of dairy products and margarine/butter/cream was also correlated with DNA damage at study end. These results are discussed in Bishop et al. (2015). No significant relationships were noted between DNA damage and vegetable, fruit, or pomegranate intake.

An inverse association between DNA damage and vitamin C intake was apparent. This was weak at baseline (p = 0.098), and became significant at study end (p = 0.007). Dietary folate intake at three months was inversely associated with hydrogen peroxide-induced DNA damage (p = 0.023). Vitamin E intake, which increased significantly (Table 4), was inversely associated with both basal and peroxide-induced DNA damage at the end of the study. However, with p-values of 0.175 for each, these did not attain statistical significance.

There were no significant relationships between C-reactive protein, PSA and DNA damage. A non-significant trend towards a correlation between C-reactive protein and peroxide-induced DNA damage was observed (p = 0.156 and p = 0.223 at baseline and three months, respectively).

Discussion

The primary goal of this pilot study was to establish both feasibility and likelihood of benefit, as determined by a reduction in inflammation and DNA damage, for New Zealand men with prostate cancer following a modified Mediterranean diet. We sought to enrol men with untreated prostate cancer; however, due to low numbers of volunteers, men with low Gleason (3 + 3 or 3 + 4) who had previously had treatment were included. It was anticipated that as our subjects had volunteered to participate in this intervention, they would be motivated, and thus amenable to changes that might be seen as advantageous in delaying disease progression. This was observed, and is consistent with what has been previously demonstrated in regards to cancer, motivation and dietary changes (Allen et al., 2008; Avery et al., 2013). It was anticipated that the dietary intervention would have demonstrable effects on DNA damage, PSA and C-reactive protein.

Adherence to a low inflammatory diet such as that used in this study may help to mitigate inflammation-associated increases in oxidative stress, genomic instability and damage to DNA (Kundu & Surh, 2012).

Inflammation was evaluated by measuring C-reactive protein, a non-specific acute phase protein that serves as a surrogate for interleukin 6, an inflammatory cytokine associated with angiogenesis, tumour growth and metastases. Within 6 h of an inflammatory assault, C-reactive protein increases. Constant levels are maintained commensurate with inflammatory processes and rapid clearance occurs when the stimulus is removed (Pepys & Hirschfield, 2003). No change in C-reactive protein was observed in our cohort after three months of the dietary intervention, which is in contrast to other studies using a similar dietary regimen (Marlow et al., 2013; Ellett et al., 2013). Noteworthy in this regard is that baseline C-reactive protein in this group was low, with 95% of the cohort within the normal range of <5 mg/L, leaving little room for improvement. An inverse association between C-reactive protein and physical activity levels that was noted at the outset of this study was not evident at study end. Medication did not change in the group, with the exception of one participant who ceased taking non-steroidal anti-inflammatory drugs during the study period. There was no resultant increase in C-reactive protein in this participant.

The effect of excess adiposity is an important consideration on the results seen. Obesity is a chronic, low-grade inflammatory state, which has been associated with both incidence and progression of prostate cancer (Ho et al., 2012). It was expected that study participants, four of whom were obese at the beginning of the study, might lose weight by following the dietary recommendations (Martinez-Gonzalez et al., 2012). This did occur (mean weight loss 2.3 kg), and while desirable in overweight men, may have masked anti-inflammatory attributes of the diet. When stored adipose tissue is catabolised, the pro-inflammatory omega-6 fatty acid, arachidonic acid, tends to be liberated (Phinney et al., 1991). Thus true benefit in terms of lowering of inflammation might be better observed once body weight has stabilised. This would be best assessed following a longer intervention. No significant associations between body weight, BMI (or changes thereof) and DNA damage were noted in this study. C-reactive protein responds to a number of factors. Participants were not evaluated for injury or opportunistic infection at either the beginning or end of the study. While men retrospectively reported that they were “well” at the time of both blood draws, minor injuries or low-grade infections have the potential to increase acute phase inflammatory markers, C-reactive protein included. Similarly, study participants were not assessed for other factors that might impact C-reactive protein, such as sleep disturbances (Meier-Ewert et al., 2004) and food intake the day of sample collection (Margioris, 2009; Farnetti et al., 2011).

Levels of PSA were largely unchanged over the course of the study. However, PSA was below the level of detection in the majority of study participants, as is associated with successful treatment of prostate cancer. The lack of change in PSA, particularly in those men who had not undergone treatment for prostate cancer (n = 6), may indicate benefit in terms of PSA doubling time. While this is best evaluated over a longer period of time, mitigation of PSA increases have been demonstrated after 3 months of a dietary shift towards a more plant-based pattern (Saxe et al., 2008).

There are many individual components of the Mediterranean diet that have been studied in regards to their effect on a number of health outcomes. The health advantages of a diet that is high in fruit and vegetables, ostensibly due to the diversity of nutrients, with high levels of antioxidants and fibre associated with such dietary patterns is generally accepted.Indeed, lower levels of inflammation and increases in antioxidants have been correlated to fruit and vegetable intake (Root et al., 2012). While not all studies concur (Ambrosini et al., 2007; Boffetta et al., 2010), there is evidence suggesting benefit from vegetable and fruit intake in regards to prostate cancer (Riso et al., 1999; Hardin, Cheng & Witte, 2011; Shahar et al., 2011).

The main benefit (weight loss aside) associated with this dietary intervention was reduction in DNA damage after three months, when compared to baseline data. This outcome was inversely associated with dietary adherence (p = 0.013).

Three months is considered sufficient time to determine the impact of diet on DNA repair in lymphocytes. As part of the circulatory system lymphocytes are constantly exposed to the positive and negative effects of diet and lifestyle. Therefore, they are an ideal target cell to assess the nutritional or chemical effect on DNA damage, regardless of their age. Other studies have demonstrated the effect of dietary on DNA repair in lymphocytes in as little as 21 (Riso et al., 1999) and 24 days (Guarnieri et al., 2008).

DNA damage has been positively associated with prostate cancer risk (Lockett et al., 2006), hence increased DNA protection and repair is a highly desirable outcome, further supporting the benefit of a diet high in antioxidants and low in saturated fat. Specific foods and nutrients, particularly antioxidants and polyphenol compounds, can positively affect DNA repair (Duthie et al., 1996; Giovannelli et al., 2002; Machowetz et al., 2007; Guarnieri et al., 2008). For example, consumption of green tea (Kanwal et al., 2014), broccoli (Riso et al., 2010) and vitamin C intake (Fraga et al., 1991) have been associated with increased DNA repair in previous studies, while increases in DNA damage have been attributed to oxidative stress (Freitas et al., 2012; Kundu & Surh, 2012) and peroxidation of fatty acids (Gropper, Stepnick & Smith, 2013). The benefit of a Mediterranean diet on markers of DNA damage has been reported in women with the metabolic syndrome (Mitjavila et al., 2013). As far as the authors are aware, this has not previously been reported in men with prostate cancer.

Diet is a complex interaction of a wide range of foods and numerous individual compounds. Genetic and epigenetic modifications can be affected by dietary phytonutrients, which modulate DNA methylation and may induce or enhance DNA repair, the isolation of these compounds is a commonly used, but reductionist approach, to nutritional research. In this study, dietary adherence scores informed a comparison of the adoption of the diet as a whole, as well as the integration of various aspects of it. The aspects of this diet that study participants found the most acceptable were the incorporation of whole grains, olive oil, pomegranate juice, substitution of red meat for chicken, and reducing consumption of sweetened beverages. Each of these achieved >85% compliance overall. On the other hand, the least embraced components were inclusion of sofrito, green tea, vegetables, broccoli, and adoption of recommended guidelines for alcohol (in particular red wine). While compliance on these latter items was less than 60%, changes in most did reach statistical significance (Tables 4, 6 and 7).

It is apparent from the inverse association of adherence to the recommended diet with DNA damage that this overall dietary pattern could be of benefit for men with prostate cancer. While the possibility exists that reduced damage to DNA may be attributable to other characteristics of this cohort, no parameter reported here suggests that this was the case. Other factors associated with the Mediterranean diet, when compared to a typical western diet, include alterations in the type of dietary fat and resultant changes in blood fatty acids. Such changes did occur in this study, as is evident by the decrease in energy from saturated fat (Fig. 2). Relationships between DNA damage, dietary fats and blood fatty acids are explored elsewhere (Bishop et al., 2015).

An inquiry into relationships between individual food items and the benefits seen on this diet aids in justification of continued inclusion or otherwise in future studies. Such data is particularly useful when recommending the incorporation of foods that may be considered unusual and to support the development of strategies to aid increased compliance with less accepted recommendations.

While participants indicated that they were more inclined to choose chicken over red meat, overall complicity to the criteria for restriction of red meat consumption (to less than once weekly) did not indicate that this had actually occurred. In general, New Zealand consumption of beef and lamb is high (FAO, 2013), which is reflected in the baseline data. At the beginning of the study participants were consuming almost 4 servings of red and/or processed meat per week. Hence, while the reduction noted was significant (p < 0.001), intake was still higher than requested, suggesting that minimising red meat was a difficult change for participants to make. This may have been caused or compounded by the fact that this study was conducted over the summer period. The New Zealand summer coincides with the festive season during which barbeques, as social events, are a common aspect of the culture.Thus avoidance of red meat may have been challenging for some participants. If indeed this was a major factor, it also raises the possibility that consumption of pro-carcinogenic heterocyclic amines, as is associated with meat cooked at high temperatures (Norrish et al., 1999), may have negated some of the benefits of the overall diet.

Green tea was not well accepted, with palatability being the reported obstacle. Nevertheless, green tea intake was associated with significant reduction in DNA damage (p = 0.002). This is in alignment with the remarkable antioxidant and anti-inflammatory properties associated with green tea, which contribute to its effect on DNA methylation (Roda & Eubank, 2012). Green tea consumption was reported by 25% of participants at baseline and 60% at study end. However, less than half were consuming the recommended intake of 2 cups daily. This relatively low uptake, but notable benefit, suggests that the incorporation of a beneficial dietary component may result in substantial gains, even if target levels are not attained.

No associations were noted for broccoli intake, which is in contrast to other reports (Latté, Appel & Lampen, 2011; Ferguson & Schlothauer, 2012). However, the inverse correlation noted between folate intake and DNA damage is consistent with current understanding of the role of folate in DNA methylation (Gropper, Stepnick & Smith, 2013), and concurs with a recent study in which folate’s role in DNA stability was demonstrated (Ong, Moreno & Ross, 2011). In the current study, folate intake was calculated from diet diaries, and broccoli intake was determined from an adherence questionnaire, which was not sensitive to other sources of either folate or other cruciferous vegetables.

The role of vitamin C in regards to cancer is conflicting (Fraga et al., 1991; Herbert et al., 2006) making the finding of this study difficult to compare with other research. A dose–response relationship has been observed in regards to protection from DNA damage, with increasing vitamin C benefiting those with low baseline levels (Herbert et al., 2006; Freitas et al., 2012). In our study a significant inverse association was apparent between vitamin C intake and DNA damage (p = 0.007) at study end. Dietary vitamin C in the cohort was not low (Table 6). Examination of the data indicated that the source of this water soluble antioxidant shifted from fruit at baseline to vegetables at the end of the study. However, overall vegetable intake did not change. Furthermore, vegetables are more likely to be consumed cooked, which reduces vitamin C content (Gropper, Stepnick & Smith, 2013). Together, this suggests that vitamin C was not solely associated with the effect noted, but rather, that a nutrient–nutrient interaction, such as with vitamin E, may have occurred. Vitamin E intake increased over the course of the study (p = 0.005) (Table 6).This is congruent with current understanding of the synergistic relationship that exists between these two antioxidant vitamins (Gropper, Stepnick & Smith, 2013). A large prospective study by Wright et al. (2007) suggested that dietary (but not supplemental) vitamin E may be an important nutrient in reducing the risk of advanced prostate cancer.

Diet is a complex interaction of phytonutrients and it is an accumulative and/or synergistic effect of these that confers overall benefit. In other words, overall dietary patterns are possibly more important than individual components—a concept that has been proposed elsewhere (Viscogliosi et al., 2013).

From the results, it appears that the adoption of such a dietary pattern is feasible in motivated men. Not only were the changes, for the most part, embraced, but feedback from individual participants was encouragingly positive. While not part of the study design, at the end of the study and 3 months later, informal feedback was gathered from the participants. The information provided indicated that participants were heartened by their weight loss and the majority continued with the new diet. Study participants also reported improvements in overall energy and well-being, as well as positive effects on a range of other factors, including serum cholesterol, arthritic pain, and nocturia. Formal collection and evaluation of such data in future studies would contribute to strategies for supporting dietary change in men with prostate cancer.

The ethnic homogeneity of the group in the current study meant that the influence of variances in genetic expression was reduced. However, a number of other confounders, including genetic influences, were not controlled for as this would have rendered the group too small for meaningful analysis. The authors recognise the importance of gene-diet interactions and acknowledge that any testing related to genetic expression requires a larger study to demonstrate relevance. Despite the fact that this was a small feasibility study, positive outcomes were noted within a short time frame. In order to apply the findings to a general population, it is necessary to confirm the results with a larger cohort. Such a study would allow stratification to control for a variety of confounding factors, such asadiposity, baseline dietary patterns, co-morbidities and lifestyle habits. The inclusion of a control group of men without prostate cancer would also be advantageous.

Conclusions

We demonstrated that dietary change towards a Mediterranean-style pattern is both achievable and beneficial for men with prostate cancer in New Zealand, albeit in a small and motivated group. While there have been numerous studies evaluating the impact of a Mediterranean diet on clinical indicators of disease (Giovannelli et al., 2002; Menendez & Lupu, 2006; Fernández-Real et al., 2012; Kontogianni et al., 2012; Marlow et al., 2013; Mitjavila et al., 2013; Viscogliosi et al., 2013; Ellett et al., 2013) including prostate cancer (Ferrís-Tortajada et al., 2012; Kenfield et al., 2014), we believe this is the first time such a study has been conducted in regards to prostate cancer in New Zealand. This pilotstudy shows that a holistic approach to diet may contribute to modulation of DNA damage in spite of low baseline levels of inflammation. This is quite possibly due to an undeterminable synergistic effect of dietary components and associated phytonutrients.

Reduction in DNA damage was significantly associated with overall conformity to the general dietary pattern as well as intake of green tea, legumes, dietary vitamin C and folate. While no effect on inflammatory markers was demonstrated, baseline inflammation in this small cohort was low. Nonetheless, the results obtained add weight to the notion that a low-inflammatory, high antioxidant diet may be of benefit for men with prostate cancer.

Certain aspects of the diet were more acceptable to participants than others. An exploration of the challenges faced in integration of specific dietary components would inform strategies to encourage ongoing compliance, and ultimately, long-term benefit for men with prostate cancer, including those with more advanced disease.

Supplemental Information

Supplemental Information 1 DIP study data file

Click here for additional data file.

We would like to thank our New Zealand sponsors, Oil Seed Extractions Ltd., and Aoraki Smokehouse Salmon, Life Juices (New Zealand) Ltd., and Delmaine Fine Foods for donations of food.

We thank Shuotun Zhu, who conducted the phlebotomy, and Amalini Jesuthasan, who processed the laboratory samples.

Our gratitude is also extended to the men that committed themselves to this project, and their families.

Additional Information and Declarations

Competing Interests

Author Contributions

Human Ethics

Lynnette R. Ferguson is an Academic Editor for PeerJ.

Sharon Erdrich conceived and designed the experiments, performed the experiments, analyzed the data, wrote the paper, prepared figures and/or tables, reviewed drafts of the paper.

Karen S. Bishop conceived and designed the experiments, performed the experiments, reviewed drafts of the paper.

Nishi Karunasinghe contributed reagents/materials/analysis tools, reviewed drafts of the paper.

Dug Yeo Han analyzed the data.

Lynnette R. Ferguson reviewed drafts of the paper.

The following information was supplied relating to ethical approvals (i.e., approving body and any reference numbers):

Written approval provided by the Northern Y Regional Ethics Committee (New Zealand), study reference NTY/11/11/109.

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
