# Peer review of "A pilot study to investigate if New Zealand men with prostate cancer benefit from a Mediterranean-style diet"

_PeerJ, doi:10.7717/peerj.1080_

## Round 0.1 · original submission · Major Revisions

· Academic Editor

Major Revisions

A close evaluation of the comments of the reviewers to your submission, as well as my own assessment, has led to the decision that major revisions are required. Considerably more detail is required in the Materials and Methods section on patient selection/rejection, their medical profiles and details on stage of the disease; also on the quality and sourcing of the dietary supplements. The tables in the Results section also suggests compliance was not particularly high. Also, a discussion of confounding factors such as statin-use (strong effects on blood lipid profiles) and the occasional barbecue (direct effects on DNA and inflammation - even if it is only vegetables that are grilled) and the very small sample size and short duration of study should be discussed more openly.

Reviewer 1 ·

Basic reporting

Comments:
1. Title: The fact that this is a pilot study should be stated in the title
2. Abstract: Missing word “evaluated at baseline and after three months”
3. Abstract: I prefer the word suggested over demonstrated- “This small study suggested…”
4. Introduction: There is a typo at the bottom of the first page “…a low inflammatory diet.”
5. Methods: The first mention that this is a pilot study occurs in the Discussion. Given the sample size and the relatively short follow-up period, I consider this to be a pilot study. This should be mentioned in the abstract and methods as well.
6. Methods: Overall comment on this section is that it should provide more detailed information. Since the objective is to publish information that can be used in a larger study, then detailed information about the study sample, data collection tools and the intervention are needed. Specific issues include:
a. Study sample: Please provide additional information describing the sample. A previous study using the same pool of subjects is mentioned; if it is published, please provide a reference. Why were men with these Gleason scores selected?
b. Dietary recommendations: The authors state that the diet is Mediterranean style, and that it is modified to include fish, tea, etc. A detailed description of the specific recommendations issued to the subjects is needed (eg which food items specifically? how many servings? Serving sizes? Etc). This will form the basis for clearer comparison with future studies on this issue.
c. Dietary intervention: What was the intervention? Who provided the counselling? What was the content of the counselling? How often? What resources were provided (written documents, websites, etc). If the men were also supplied with food, this should be more clearly stated. How was this arranged? Did all the men accept and receive their items?
d. Outcome measures: How exactly was dietary adherence defined and measured? How was adherence determined? Was a questionnaire developed for this study or was a pre-existing one used? For any pre-existing tools or concepts please provide references. Otherwise, a description of how these tools were developed should be added.
7. Results: The sentence “No relationship was seen between sugar intake and C-reactive protein” seems out of place. This section focuses on dietary changes, not on associations with clinical parameters, which are presented later in the text.
8. Discussion: It is important to consider how certain baseline factors might affect the impact of the intervention. I believe that if a larger study is undertaken, there should be some stratified analyses focusing, for instance, on baseline BMI and dietary profile. For analyses assessing the impact on biochemical indicators, it would be useful to evaluate if individuals who are obese at baseline, for example, have a different response to the diet as compared to those who were not obese. The same might apply for the diet- those whose diet is very far from ideal might have a different response as compared to those whose diet more closely resembles the recommended diet.
9. Discussion: The authors mention that three months is sufficient time to determine the effect of diet on DNA damage. But what about the other indicators? Is three months enough time to observe a change in PSA or C-reactive protein?
10. Discussion: In a future study, it could be useful to see how changes in diet influence the clinical indicators among men without prostate cancer as well. Also, assessing the impact of dietary changes among men with more advanced prostate cancer would be of interest.
11. Discussion: Could the weight loss, or other lifestyle changes that probably accompanied the change in diet, and not the diet itself, be the driving force underlying the changes in parameters?
12. Discussion: Last sentence should say “it is” and not “it may”. “In order to apply the findings to a general population, it may first be necessary to confirm the results with a larger cohort”. This is a pilot study, so the results absolutely need to be replicated!
Overall assessment:
This is a very interesting study, but there are some important things to address. The manuscript should be re-organized to reflect the fact that this is a pilot study. The objective therefore should be to document and pilot-test, in a small sample, the various methods and measures that will be used in a larger study. The goal of the paper should be to present some preliminary findings, but more importantly, to present the methodology used in detail so that it can be replicated in larger studies, or form a sound basis for comparison with future studies. While I found the Introduction and Discussion interesting, these sections were long, and could be cut if needed. The Methods section needs to be considerably more detailed. Details to add include: study sample and recruitment, the specific dietary recommendations given, how the intervention was provided, how adherence was defined and quantified, as well as how the questionnaires were selected or developed. In addition, for a pilot study, assessment of feasibility is often carried out. Feasibility is briefly mentioned in the discussion but otherwise it isn’t really dealt with. In the Discussion the authors mention that the changes were “embraced” and “feedback was positive” but where is the data to support that? In my opinion, at the end of the 3 month intervention, data should have been collected to show how the intervention was received and this should be presented.

Experimental design

No comments.

Validity of the findings

No comments.

Reviewer 2 ·

Basic reporting

.

Experimental design

.

Validity of the findings

.

Additional comments

This manuscript describes a small observational, prospective dietary intervention study of New Zealand volunteer’s men with prostate cancer. The study investigates the benefit of three months of adherence to a Mediterranean-style dietary pattern, with specific modifications on DNA damage and inflammation.

My main criticism is the assumption by the authors that the observed difference is caused solely by a short period of adherence to a Mediterranean-style dietary pattern.

Weaknesses:
1. The observed decrease in DNA damage could also be related to other characteristics of the men
2. Diet needs to be assessed with a food frequency questionnaire administered prior to cancer diagnosis and again at three months of follow-up after the diagnosis. Moreover, adherence to the Mediterranean diet also needs to be evaluated (authors include data without describing the methodology).
3. Very small sample size and follow up period
4. A lot of important information is missing. For example:

- When were participant diagnosed with prostate cancer?
- Tumor characteristics (only Gleason score)
- Meaning of “stable prostate cancer”
- Exclusion criteria: were individuals with a previous history of cancer excluded?
- How were total energy and nutrient intakes calculated?
- The dietary intervention needs to be more explained in greater depth.
- Mediterranean-style diet definition seems mostly defined based on: olive oil (polyphenol level 233 mg/kg), salmon (200 g/week), pomegranate juice (1 L per week) and canned legumes.
- How was the participants’ degree of adherence to the Mediterranean –style diet evaluated?
- Covariates or factors potentially associated with Mediterranean diet and DNA damage and inflammation need to be taken into account.

---

## Round 0.2 · Minor Revisions

· Academic Editor

Minor Revisions

Dear authors, the manuscript has been vastly improved. However, please address the minor revisions suggested by reviewer #1.. We look forward to seeing your final submission.

Sincerely,
Thomas Sanderson, editor PeerJ

Reviewer 1 ·

Basic reporting

No comments

Experimental design

No comments

Validity of the findings

No comments

Additional comments

MANUSCRIPT REVIEW: REVISION

Journal: PeerJ
Revised Title: A pilot study to investigate if New Zealand men with prostate cancer benefit from a Mediterranean-style diet
Date: April 30, 2015

Overall assessment:
The authors have responded to each of the comments, and the manuscript is much improved. There are a few outstanding issues:
1. There are some general issues with formatting throughout the paper (line spacing and punctuation). This should be addressed prior to publication.
2. At almost three pages I still find the Introduction long, however if the editors are comfortable with it, then there is no reason to cut.
3. The manuscript has been modified throughout to reflect that this is a pilot/feasibility study which is an important change. I would suggest adding that to the sentence about the objectives, in the last line of the Introduction. I would add that this is a pilot study, and that the study was done to assess not just the benefit of the program over three months, but also its feasibility.
4. Throughout the manuscript, it seems to me that the word “serves” is sometimes used instead of “servings”. As in line 322 “4 serves of red meat”. This also happens in Table 4. The manuscript and tables should be checked for this.
5. Table 1 is unclear. First, green tea is in the avoid column, which I assume is a mistake. Fruit is also in the avoid column, is this correct? Is it because it should be limited to 2 portions? What about eggs, is this correct? Please review the contents of the columns to make sure the items are correct. Also, what does “preferred item” mean?
6. I find the formatting of Table 3 a bit awkward, I would prefer that the table is presented as one column, and the subgroups under each characteristic be indented.
7. For Table 4, I would put the * beside “Criteria for one point”.

Reviewer 2 ·

Basic reporting

The authors have addressed the points I made in my review and I have no further comments

Experimental design

No comments.

Validity of the findings

No comments.

Additional comments

The authors have addressed the points I made in my review and I have no further comments

---

## Round 0.3 · accepted · Accept

· Academic Editor

Accept

The final minor revisions that were requested have been addressed.